# Effects of an obesogenic diet on the oviduct depend on the duration of feeding

**Kerlijne Moorkens**[1]*, **Jo L. M. R. Leroy**[1], **Sara Verheyen**[1], **Waleed F. A. Marei**[1,2]

**1** Department of Veterinary Sciences, Gamete Research Centre, Laboratory for Veterinary Physiology and Biochemistry, University of Antwerp, Wilrijk, Antwerp, Belgium, **2** Department of Theriogenology, Faculty of Veterinary Medicine, Cairo University, Giza, Egypt

* Kerlijne.moorkens@uantwerpen.be

**Data Availability Statement:** All serum cholesterol, serum cytokine and OEC gene expression raw data files are available from the Mendeley database (Moorkens, K. (2022), "PONE-D-22-12367", Mendeley Data, V1, doi: 10.17632/msjbsp9t5h.1.)

## Abstract

### Research question

How long does it take for an obesogenic (high-fat/high-sugar, HF/HS) diet to influence the oviductal microenvironment? What are the affected cellular pathways and are they dependent on the genetic background of the mouse model?

### Design

Female Swiss (outbred) and C57BL/6N (B6, inbred) mice were fed either a control (10% fat) or HF/HS (60% fat, 20% fructose) diet. Body weight was measured weekly. Mice were sacrificed at 3 days (3d), 1 week (1w), 4w, 8w, 12w and 16w on the diet (n = 5 per treatment per time point). Total cholesterol concentrations and inflammatory cytokines were measured in serum. Oviductal epithelial cells (OECs) were used to study the expression of genes involved in (mitochondrial) oxidative stress (OS), endoplasmic reticulum (ER) stress and inflammation using qPCR.

### Results

Body weight and blood cholesterol increased significantly in the HF/HS mice in both strains compared to controls. In Swiss mice, HF/HS diet acutely increased ER-stress and OS-related genes in the OECs already after 3d. Subsequently, mitochondrial and cytoplasmic antioxidants were upregulated and ER-stress was alleviated at 1w. After 4-8w (mid-phase), the expression of ER-stress and OS-related genes was increased again and persisted throughout the late-phase (12-16w). Serum inflammatory cytokines and inflammatory marker-gene expression in the OECs were increased only in the late-phase. Some of the OEC stress responses were stronger or earlier in the B6.

### Conclusions

OECs are sensitive to an obesogenic diet and may exhibit acute stress responses already after a few days of feeding. This may impact the oviductal microenvironment and contribute to diet-induced subfertility.

**Funding:** This research was funded by the special research fund University of Antwerp (grant no. 36934, recipient JLMRL, https://www.uantwerpen.be/en/research/management/funding/internal-funding/university-research-bof/) and by FWO (Fonds Wetenschappelijk Onderzoek) (project no. G038619N, recipient JLMRL, https://www.fwo.be/). The funders had no role in study design, data collection and analysis, decision to publish, or preparation of the manuscript.

**Competing interests:** The authors fully declare that they have no financial or other potential competing interests.

**Abbreviations:** OECs, oviductal epithelial cells; OS, Oxidative stress.

## Introduction

The prevalence of obesity and type-II diabetes is rapidly increasing worldwide [1, 2] due to the increased consumption of the obesogenic diet and the lack of physical activity [3–5]. Metabolic syndrome has become a major health hazard of the modern world with an average prevalence of 31% [4, 6]. Recently there is increasing evidence that metabolic disorders linked to obesity can have direct negative effects on fertility [7]. Obese or overweight women are less likely to achieve spontaneous pregnancy and have lower reproductive outcomes [8]. The underlying mechanisms are progressively unravelled but are not fully elucidated.

Obesogenic diets are characterized by a high fat content [9], especially saturated fats, which are known to lead to increased adiposity, hyperlipidemia and inflammation. This results in cellular dysfunction due to lipotoxicity, reduced insulin sensitivity and oxidative stress (**OS**) [6, 10–13]. The high sugar content in these diets [9] further exacerbates the severity of this metabolic disorder [14].

High-fat/high-sugar (**HF/HS**) diet-induced obesity in mice has been used in many studies to mimic and understand the mechanisms by which an obesogenic diet impacts fertility. Beside possible systemic endocrine disruption, the impact of HF/HS diet and obesity has been shown to be mainly mediated through direct detrimental effects on the reproductive organs during oocyte and embryo development and during pregnancy [15]. The direct impact of obesity on the ovarian follicular fluid (**FF**) microenvironment and oocyte quality is relatively more documented and clearly elucidated in animal models [16–23] and women [24–26]. Oocytes matured under lipotoxic conditions exhibit altered mitochondrial functions and OS, an aberrant proteomic profile and metabolic activity, ultimately leading to lower oocyte developmental competence and lower embryo quality [16–18, 21, 27, 28].

The effects of maternal metabolic stress on the oviduct are much less well characterized. This is however important as several crucial reproductive events take place in the oviduct such as oocyte and sperm capacitation and fertilization, syngamy, embryo genome activation and epigenetic (re)programming [29–32]. The oviductal secretory epithelial cells and the oviductal fluid provide nutrients, growth factors and antioxidants (AO) to support early embryo development (which is in most mammalian species during the first 3–5 days post-fertilization) and changes in this micro-environment may lead to lower pregnancy success or even to offspring health defects [33, 34].

Only a few studies have previously expanded on that. For example it has been shown that cows suffering from severe negative energy balance and systemic hyperlipidemia and lipotoxicity due to fat mobilization exhibited reduced oviductal ability to support embryo development after embryo transfer [35, 36]. Additionally, Jordaens et al. (2020) showed that *in vitro* exposure of bovine oviductal cells to elevated non-esterified fatty acid (NEFA) concentrations leads to direct embryo toxicity and to a reduced oviductal ability to support and protect early embryo development [37]. Nevertheless, potential direct effects of HF/HS diet-induced obesity on the oviductal microenvironment have not been investigated in humans or in relevant animal models.

The direct impact of a HF/HS diet and obesity on cellular functions depends on the duration of the direct exposure to the dietary components. Systemic inflammation was reported in mice as early as 3 days (3d) after the start of the high-fat diet (HFD) and was shown to be limited to distinct phases (acute and late periods) with an adaptive period in between [38]. In contrast, other studies demonstrated a gradual increase [39] or even no inflammatory responses [3] in mice fed a HFD. Local inflammatory responses are also expected at the cellular level and may contribute to reproductive dysfunction [40, 41]. On the other hand, diet-induced hyperlipidemia is also known to induce lipotoxicity in non-adipose tissues, alter cellular metabolism

due to mitochondrial dysfunction, and increase OS levels [11, 13, 42]. Nevertheless, it is not known if the microenvironment of the oviduct exhibits such pathophysiological changes in response to an obesogenic diet. Also the timeframe during which these alterations may develop after consuming an obesogenic diet has not been previously investigated. Fundamental insight in this area is necessary to determine the potential impact on embryo development and the timeframe at which the early embryo might be at risk.

We have recently shown that the sensitivity to HFD-induced obesity and its influence on the metabolic profile and oocyte quality was dependent on the mitochondrial and/or genetic background of the mouse model. This was demonstrated by comparing the outbred Swiss mice with the inbred C57BL/6N mice [17]. Such difference may also influence the metabolic impact on the oviductal microenvironment.

In this study, we hypothesized that feeding a HF/HS diet can lead to acute and/or long-term changes in the oviduct of mice, particularly related to oxidative and cellular stress levels and inflammatory responses and that the nature and magnitude of these changes depend on the duration of feeding. This may also be dependent on the mouse strain used in the model. To test this hypothesis, we aimed to analyze serum and oviductal epithelial cell (**OEC**) samples collected from Swiss and C57BL/6N mice that were exposed to a HF/HS diet for different periods of time, namely 24 hours, 3 days, 1 week, 4 weeks, 8 weeks, 12 weeks and 16 weeks. Outcome parameters were focused on serum total cholesterol and cytokine concentrations and on the expression of genes involved in response to OS, endoplasmic reticulum (ER) stress and inflammation in OECs.

## Material and methods

### Ethical approval

All procedures in this study were approved by the ethical committee of the University of Antwerp and performed according (ECD approval number nr. 2014–57). All methods were performed in accordance with the relevant guidelines and regulations. This study complies with the ARRIVE guidelines. Euthanasia was done by decapitation.

### Experimental animals, diet and experimental design

Five-week-old, non-pregnant, sexually mature female outbred Rj:Orl Swiss mice (hereafter referred to as "Swiss" mice, n = 70, Janvier labs), and inbred C57BL/6N mice (hereafter referred to as "Black 6" (B6) mice, n = 70, Janvier labs) were used.

Mice of each strain were randomly divided into two groups with ad libitum access to either a control diet (CTRL) or a high-fat/high-sugar diet (HF/HS). A HF/HS diet is more representative for the typical Western style diet, compared to high-fat diet only. The HF/HS group was fed with 60 kJ% fat (beef tallow) and 9.4% sucrose diet (E15741-34, Sniff diets, Soest, Germany) in combination with supplementing drinking water with fructose at a final concentration of 20% (Merck, 102109450). Beef tallow was chosen over lard because beef tallow contains more saturated fatty acids. Fructose was chosen over glucose or sucrose since a high consumption of fructose has proven to lead to insulin resistance and obesity in rodents [43]. A high-fat/high-fructose diet is known to induce metabolic syndrome and type 2 diabetes in mice [9, 44]. Fructose will also stimulate higher food consumption because it is more palatable and has a reduced capacity to stimulate satiety. Furthermore, fructose appears to be a better inducer of metabolic syndrome compared to glucose [45].

Mice in the control group had ad libitum access to water and were exposed to a matched, purified (not a grain-based chow diet) CTRL diet (E157453-04, Sniff diets, Soest, Germany),

containing 10 kJ% fat and 7% sucrose. This diet is also lower in saturated fatty acids. Food intake of all mice was monitored and mice were weighed weekly during the whole experiment.

Mice were euthanized by decapitation at 7 time points (n = 5 per treatment per time point): 24 hours, 3 days, 1 week (1w), 4w, 8w, 12w and 16w after the start of feeding of the CTRL and the HF/HS diets. Mice were not fasted before euthanasia to avoid counteracting the dietary effects especially in the early timepoints. Serum and OECs were collected as explained hereafter to study different outcome parameters. Female mice were exposed to bedding from male cages 24 hours before euthanasia to synchronize their estrous cycles (Whitten effect) and to collect the samples during the follicular phase (before ovulation), ensuring no follicular cells or oocytes are present in the oviduct at sample collection. The time points at which samples were collected are based on the results of previous studies where multiphasic acute and long-term effects of a HFD on general metabolic features have been reported [38].

## Live body weight

The weight of each mouse was recorded weekly. The weight gain data is derived from 35 mice per dietary group per strain at the first time point. The number of mice were reduced after each time point due to culling of a subset of animals (n = 5 per strain) at each time point for sample collection.

## Serum collection and analysis

Mice were decapitated, and blood was immediately collected and centrifuged after 30 min at 6000xg for 2 min at 4˚C. Serum was then stored at -80˚C. Serum samples of both Swiss and B6 mice of all time points were analyzed for total cholesterol concentrations in a commercial certified laboratory (Algemeen Medisch Labo, AML, Antwerp, Belgium) using automated facilities. Total cholesterol concentrations were measured on an Abbott Architect c16000 (Abbott, Illinois, U.S.A). Serum cytokine concentration was determined using a multiplex bead-based immunoassay according to the manufacturer's guidelines (LEGENDplex™ mouse cytokine panel 2 kit, BioLegend) and flowcytometry (FACSCantoII). The kit allows simultaneous quantification of 13 mouse cytokines (IL-1α, IL-1β, IL-6, IL-10, IL12p70, IL-17A, IL-23, IL-27, MCP-1, IFN-β, IFN-γ, TNF-α and GM-CSF). This was measured in the Swiss serum samples but not in the B6 mice due to insufficient volume of the collected serum. An overview of the different cytokines and their function can be found in S1 Table. The setup of the flow cytometer was performed according to the setup procedure provided by the manufacturer. As a validation step, a template for data acquisition was obtained using the raw beads that were provided in the kit. The Median Fluorescence Intensity (MFI) of each cytokine was measured in duplicate. The individual cytokine concentrations were calculated by linear equation using a 7 point standard curve.

## Oviductal epithelial cells collection and analysis

At each time point, mice were dissected and the oviducts were collected in L15 medium (Thermo Fisher Scientific, Belgium) supplemented with 50 IU/mL penicillin G sodium salt (Merck, Belgium). The oviducts were straightened and milked with a bent needle for cell collection (OECs) in L15 medium, which was done in a DNAse and RNAse-free environment. The medium containing cells was transferred to a vial which was then centrifuged at 600xg for 10 minutes at 25˚C and the supernatant was removed. This centrifugation step was repeated after adding RNAse-free phosphate buffered saline (PBS; Life Technologies, Belgium) for pellet washing. The supernatant was removed and the pellets were snap frozen in liquid nitrogen (LN$_2$) and stored at -80˚C.

## Quantification of gene expression using real time qPCR

**RNA extraction.** Total RNA was extracted from OECs of both Swiss mice and B6 mice (5 mice/ strain/ treatment/ time point) using a TRIzol® (Invitrogen) based protocol according to the manufacturer's instructions. RNA samples were stored at -80˚C until cDNA synthesis. RNA purity and concentrations were assessed using a NanoDrop™. All RNA samples had an acceptable 260/280 ratio. RNA integrity was also checked using an Agilent RNA 6000 nano kit and a Bioanalyzer (Agilent, Santa Clara, California, U.S.A) and all samples had acceptable RIN values ($\geq$7) except a few samples collected at 24h which were excluded from the analysis. A few other samples were not included in the analysis due to a very low RNA yield. Therefore, gene expression data is reported based on at least three samples per treatment per time point.

**cDNA synthesis.** cDNA was synthesized from 1750 ng RNA/sample using Omniscript RT kit (Qiagen). First, RNA samples were treated with DNase (Promega RQ1 RNase-free DNase kit) to eliminate genomic DNA contamination.

**Quantification of gene expression using qPCR.** Transcripts of the target genes of interest were quantified by quantitative Polymerase Chain Reaction (qPCR) (QuantStudio™ 3 Real-Time PCR System (Thermofisher Sci)) using SYBR green (SsoAdvanced Universal SYBR Green supermix, Bio-Rad). RT-negative control samples and no template control samples (lacking cDNA template) were included, and all samples were analysed in duplicates in the same run per gene. Genes of interest were related to OS (*SOD2*, *PRDX1*, *PRDX3*, *PRDX6*, *NRF1*, *NRF2*), ER-stress *(BiP*, *ATF4)*, mitochondrial stress (*HSPE1*, *HSPD1*), chaperons (*HSPA8*) and inflammation (*IL-1β*, *IL-6*, *NFkB*, *TNF-α*) (an overview of the genes of interest and their function can be found in S2 Table). Quantification (Cq values) was normalized using the geometric mean of 3 housekeeping genes (*ACTB*, *B2M*, *H2AFZ*) for the Swiss samples and 2 housekeeping genes (*ACTB*, *PPIA*) for the B6 samples, according to their stability scores calculated by the NormFinder add-in in Excel (MOMA, Aarhus University Hospital, Denmark). The relative expression of each gene was calculated using the comparative quantification cycle method ($2^{-\Delta\Delta CT}$), as described by Livak and Schmittgen (2001) [46]. Forward and reverse primers for all genes of interest were designed using the Primer-BLAST tool of the U.S. National Center for Biotechnolgy Information (NCBI) and ordered at Sigma-Aldrich. The annealing and reading temperature of all assays were optimized using a gradient qPCR on a QuantStudio™ 3 Real-Time PCR System (Thermofisher Sci). An overview of all primer sequences, expected PCR product lengths, GeneBank accession numbers and temperatures can be found in Table 1.

## Statistical analysis

Each individual mouse is considered as an experimental unit. Power analysis was performed with the PS program for Power and Sample Size calculations (version 3.1.2, 2014 (from Vanderbilt University)) to achieve a power (1-β) of 90% to detect statistical differences at P-value (*P*) <0.05 based on expected mean differences from similar previous experiments performed in our laboratory. All statistical analyses were carried out using IBM Statistics SPSS (IBM SPSS statistics version 26). All the outcome measures generated numerical data and were checked for normality of distribution and homogeneity of variance.

Live body weight was analysed using repeated measures ANOVA to study the main effects and interaction of treatment and time. In addition, two-tailed independent sample T-tests were performed to study the effect of treatment on body weight within each time point. Serum cholesterol data were analysed using a univariate ANOVA test on the control groups of different time points to check for potential aging effects. In addition, two-tailed independent sample T-tests were performed to study treatment effects within each time point.

**Table 1. Primer sequences, expected PCR product lengths, GeneBank accession numbers and temperatures of the primers used for qPCR.**

| GENE | PRIMER SEQUENCE (5'-3') | FRAGMENT SIZE (BP) | GENEBANK ACCESSION NO. | EXON SPANNING / INTRON FLANKING | ANNEALING T°C | READING T °C |
|------|------------------------|--------------------|------------------------|---------------------------------|---------------|--------------|
| **Oxidative stress** | | | | | | |
| *SOD2* | CTGGACAAACCTGAGCCCTA | 186 | NM_013671.3 | Intron | 63 | ≤77 |
| | GCAGCAATCTGTAAGCGACCT | | | | | |
| *NRF1* | CTCATCCAGGTTGGTACAGG | 164 | NM_010938.4 | Intron | 59 | ≤80 |
| | GTCGTCTGGATGGTCATTTC | | | | | |
| *PRDX1* | TGTCCCACGGAGATCATTGC | 120 | NM_011034.4 | Exon | 63 | ≤76 |
| | GGGTGTGTTAATCCATGCCAG | | | | | |
| *PRDX3* | TCGTCAAGCACCTGAGTGTC | 147 | NM_007452.2 | Exon | 63 | ≤81 |
| | TTGGCTTGATCGTAGGGGAC | | | | | |
| *PRDX6* | GTTGACTGGAAGAAGGGAGAGA | 246 | NM_007453.4 | Exon | 63 | ≤81 |
| | GCCACGATCTTTCTACGGAC | | | | | |
| *NRF2* | CTCCCAGGTTGCCCACATTC | 177 | NM_010902.4 | Exon | 63 | ≤80 |
| | GAGCTATTGAGGGACTGGGC | | | | | |
| *Endoplasmic reticulum stress* | | | | | | |
| *BiP (HSPA5)* | AGGTGGGCAAACCAAGACAT | 158 | NM_001163434.1 | Intron | 63 | 76–78 |
| | CTTTGGTTGCTTGTCGCTGG | | | | | |
| *ATF4* | CGGCTATGGATGATGGCTTG | 156 | NM_001287180.1 | Intron | 59 | ≤80 |
| | AGAGCTCATCTGGCATGGTTT | | | | | |
| *Mitochondrial stress* | | | | | | |
| *HSPE1* | GGAGGGAAAGGAAAGAGTGGAG | 211 | NM_008303.4 | Intron | 63 | ≤77 |
| | TAGTTCAGACATCAGTGGAATGGC | | | | | |
| *HSPD1* | GCCAATAACACAAACGAAGAGG | 146 | NM_010477.4 | Intron | 59 | ≤80 |
| | GCATCCACAGCCAACATCAC | | | | | |
| *Protein folding (chaperons)* | | | | | | |
| *HSPA8* | CCTCGGAAAGACCGTTACCA | 152 | NM_031165.5 | Exon | 63 | ≤80 |
| | AGCCGTAAGCAATAGCAGCA | | | | | |
| **Inflammation** | | | | | | |
| *IL-1β* | TGTCTTTCCCGTGGACCTTC | 255 | NM_008361.4 | Exon | 59 (can also be 61 or 63) | ≤80 |
| | AGCTCATGGAGAATATCACTTGTTG | | | | | |
| *IL-6* | CGTGGAAATGAGAAAAGAGTTGTG | 251 | NM_031168.2 | Exon | 61 | 78 |
| | TCTGAAGGACTCTGGCTTTGTC | | | | | |
| *NFkB* | CCTGCAACAGATGGGCTACA | 201 | NM_008689.2 | Intron | 59 | 84 |
| | TTGCGGAAGGATGTCTCCAC | | | | | |
| *TNF- α* | GTCCCCAAAGGGATGAGAAGT | 123 | NM_013693.3 | Exon | 63 | 80 |
| | TTGCTACGACGTGGGCTACA | | | | | |
| **Housekeeping genes** | | | | | | |
| *B2M* | GGTCTTTCTATATCCTGGCTCACA | 126 | NM_009735.3 | Exon | 61 | ≤79 |
| | TTGATCACATGTCTCGATCCCA | | | | | |
| *ACTB* | GCAAGTACTCTGTGTGGATCGG | 148 | NM_007393.5 | Exon | 61 | ≤82 |
| | AACGCAGCTCAGTAACAGTCC | | | | | |
| *H2AFZ* | CCTCACCGCAGAGGTACTTGA | 96 | NM_016750.3 | Exon | 63 | ≤76 |
| | CCACGTATAGCAAGCTGCAAG | | | | | |
| *PPIA* | ATGGCAAGCATGTGGTCTTTGG | 198 | NM_008907.2 | Exon | 61 | ≤80 |
| | GGGTAGGGACGCTCTCCTGA | | | | | |

Serum cytokine and gene expression data were not normally distributed. Aging effects were tested by comparing the controls of different timepoints using a Kruskal Wallis test with Bonferroni correction. Since no significant age effects could be detected, and based on preliminary comparisons between the HF/HS and control at each time point, and patterns of gene expression, it was decided to merge the data into biologically relevant distinct phases. Differences at 3d were considered as "acute phase" changes and those at 1w are referred to as an "early phase". Data from 4w and 8w were merged as a "mid phase" and those from 12w and 16w were presented as a "late phase". At each phase, cytokine data and gene expression data were analysed using two-tailed independent sample T-tests on log-transformed data to study the treatment effect.

All data are expressed as mean +/- S.E.M. (standard error of mean). Data were considered significantly different at $P \leq 0.05$ and higher P-values up to 0.1 are described as tendencies.

Pearson correlation was performed to assess the relationship between serum cholesterol concentrations, body weight and gene expression patterns in Swiss mice and B6 mice.

## Results

### The impact of the HF/HS diet on body weight

Repeated measures ANOVA showed a significant treatment x time effect for both Swiss mice and B6 mice ($P < 0.001$). HF/HS diet progressively increased body weight of both Swiss and B6 mice. The difference in body weight between the HF/HS and the control group was already significant from 1w onwards in the Swiss mice, and from 2w onwards in B6 mice. Bigger error bars, and thus more variance, was observed in the Swiss strain compared to the B6 strain. Body weight continued to increase in the B6 mice until 16w while it reached a plateau at 12w after the onset of the dietary treatment in the Swiss mice (Fig 1).

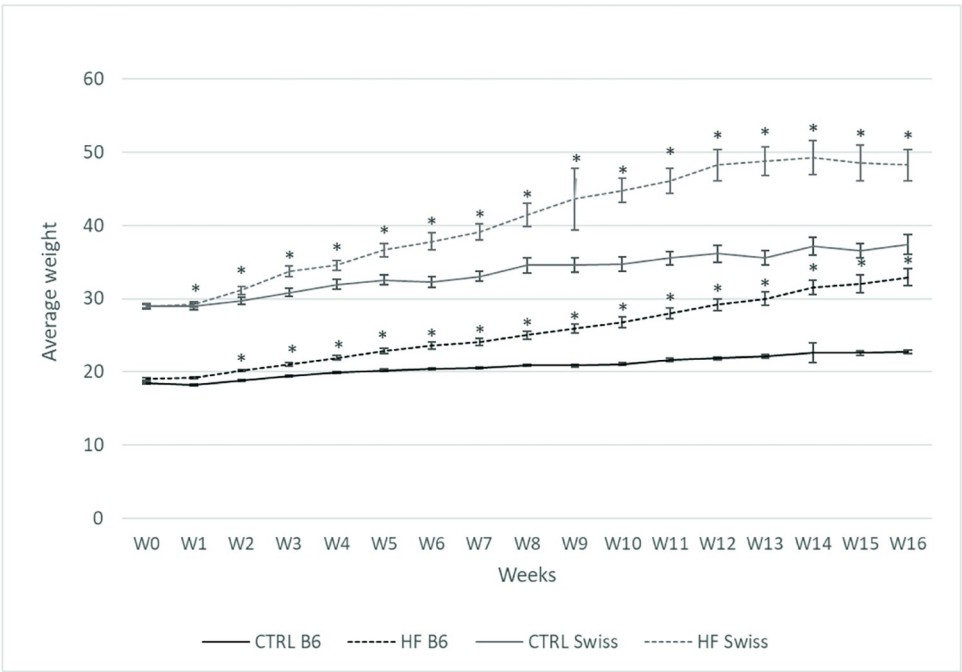

**Fig 1. The effect of HF/HS diet on body weight in Swiss mice and B6 mice.** Experimental feeding was started at W0. Body weight was weekly monitored. Data are shown as mean ± SEM. Significant difference ($P \leq 0.05$) between treatment groups within each strain per time point are indicated by an asterisk (*).

## Serum total cholesterol concentrations

Based on the two way ANOVA, no effects of age on serum total cholesterol concentrations were detected for both the Swiss ($P = 0.544$) and the B6 ($P = 0.303$) strain. In Swiss mice, there was no effect of diet on blood cholesterol concentration at 24h and 3d ($P > 0.1$). Feeding the HF/HS diet for 1 week significantly increased blood total cholesterol concentrations ($P = 0.004$). This difference could not be detected after 4 weeks ($P = 0.136$) but was more distinct again at 8w, 12w and 16 weeks ($P = 0.006$, $P = 0.068$ and $P = 0.085$ respectively) (Fig 2A). Less variation was observed in the B6 strain. Exposure to the diet for only 3d already resulted in a significant increase in blood total cholesterol concentrations ($P = 0.001$). This significant increase in cholesterol concentration remained present at all time points thereafter (1w to 12w: $P < 0.001$; 16w: $P = 0.003$) (Fig 2B).

In addition, Pearson correlation showed that serum cholesterol concentration correlated positively with body weight for both Swiss mice (Pearson correlation factor (r) = 0.322; $P = 0.008$) and B6 mice (r = 0.536; $P = 0.000$) (S3 and S4 Tables).

## Serum cytokine concentrations of Swiss mice

Based on the Kruskal Wallis test, no effects of age could be detected ($P > 0.1$) for any of the tested cytokines. The concentrations of GM-CSF, IFN-β, IL-1α, IFN-γ, MCP-1, TNF-α and IL-17A were not affected by diet ($P > 0.1$) at any of the time points (Fig 3A–3G). Whereas, the HF/HS group exhibited higher serum concentrations of IL-23 ($P = 0.017$), IL-10 ($P = 0.046$), IL-27 ($P = 0.049$), IL12p70 ($P = 0.050$) and IL-1β ($P = 0.011$) and a tendency for a higher IL-6 ($P = 0.070$) only at the late phase compared with the control mice (Fig 3H–3M).

## Gene expression patterns in OECs of Swiss mice

Kruskal Wallis comparison between the controls of different time points showed that aging did not affect gene expression patterns for any of the tested genes ($P > 0.1$). *PRDX1*, *PRDX3* and *PRDX6* are all genes involved in OS. *PRDX3* mRNA expression in HF/HS OECs was already significantly increased ($P = 0.049$) after 3d compared to the controls. No differences in expression were seen during the early and mid-phase. During the late phase, *PRDX3* expression tended to be upregulated ($P = 0.101$) in the HF/HS group. *PRDX1* only showed a tendency for upregulation ($P = 0.086$) during the early phase. *PRDX6* expression was only significantly increased during the late phase ($P = 0.030$) (Fig 4A–4C). *NRF1*, a gene involved in OS pathways, did not show any differences at any time point between controls and HF/HS (Fig 4D). *NRF2*, another gene related to OS, tended to be upregulated 3d after a HF/HS diet ($P = 0.077$) and during the mid-phase ($P = 0.057$). No differences in expression between HF/HS group and control group were observed during the early phase and late phase (Fig 4E). *SOD2*, also related to OS, showed a continuous increase in expression in the HF/HS group starting at 1w. However, during the mid-phase this increase was not significant ($P = 0.064$) (Fig 4F). An upregulation of *BiP*, a marker gene related to ER-stress, was apparent in the acute phase ($P = 0.042$), mid-phase ($P = 0.014$) and late phase ($P = 0.029$). Nevertheless, *ATF4*, the downstream transcription factor of *BiP*, showed no significant differences between controls and HF/HS ($P > 0.1$) (Fig 4G and 4H). *IL-1β* was significantly upregulated ($P = 0.050$) during the late phase in the HF/HS group (Fig 4I). Finally, markers of mitochondrial stress (*HSPE1*, *HSPD1*) and protein folding (*HSPA8*) showed no different expression levels ($P > 0.1$) at any of the time points (Fig 4J–4L). Pearson correlation showed a positive correlation between *IL-1β* and *PRDX6* (r = 0.492; $P = 0.011$) and also *NRF1* and *NRF2* correlated positively (r = 0.918; $P = 0.000$), all irrespective of time point (S3 Table).

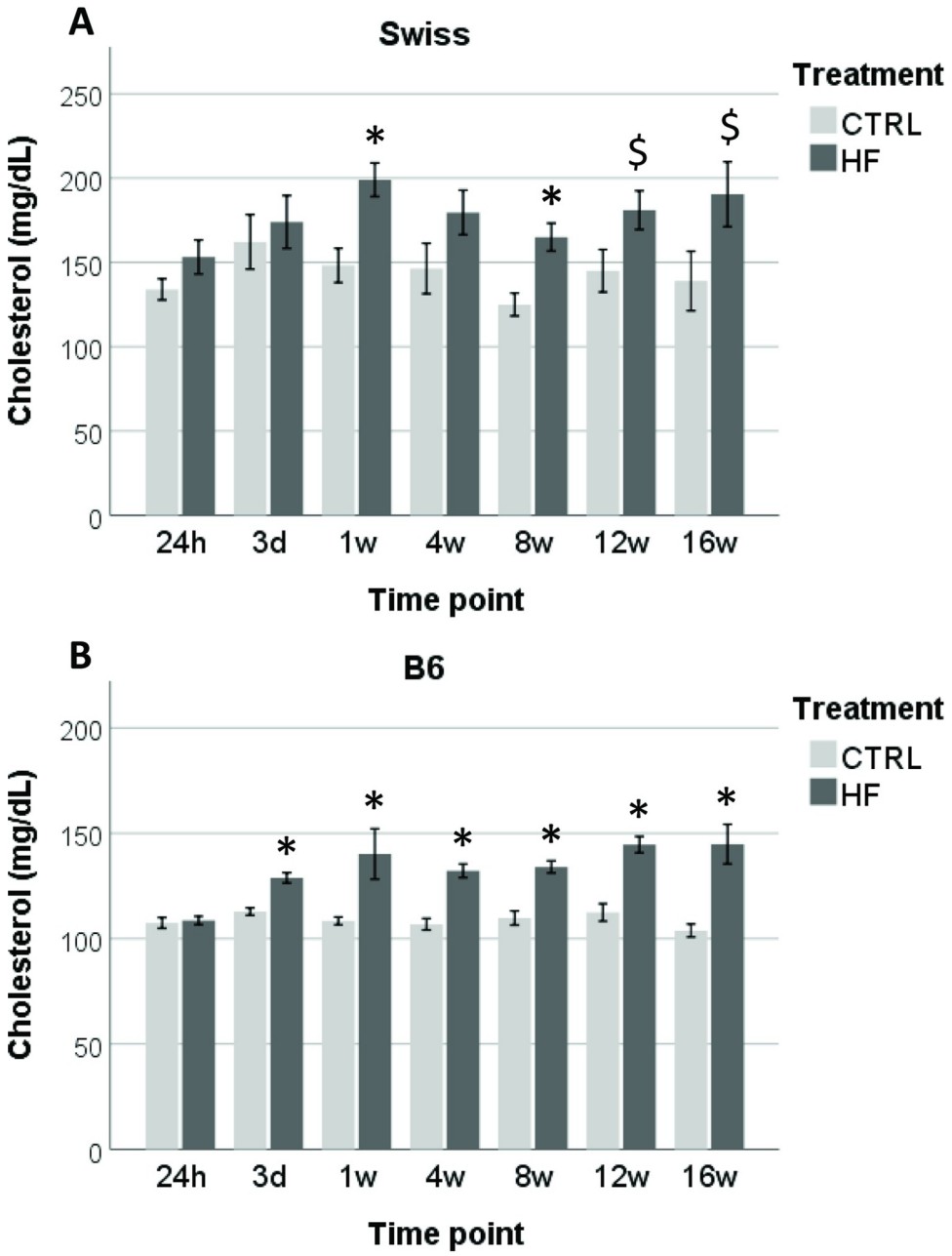

**Fig 2.** The effect of HF/HS diet on total serum cholesterol concentrations in both Swiss (A) and B6 (B) mice. Data are shown as mean ± SEM. Significant differences ($P \leq 0.05$) between HF/HS and control groups within each strain per time point are indicated by an asterisk (*). Tendencies ($P < 0.1$ and $> 0.05$) are indicated by a dollar sign ($).

## Gene expression patterns in OECs of B6 mice

Similar to the Swiss mice, no age effects could be detected for any of the tested genes ($P > 0.1$). *PRDX1*, a marker of OS, was significantly increased in the HF/HS group 3d after the start of the diet ($P = 0.019$). No differences in expression levels of *PRDX1* were measured at later time points. *PRDX3* and *PRDX6* did not show a significant difference between HF/HS and control group at any of the time points (Fig 5A–5C). No significant differences in mRNA expression

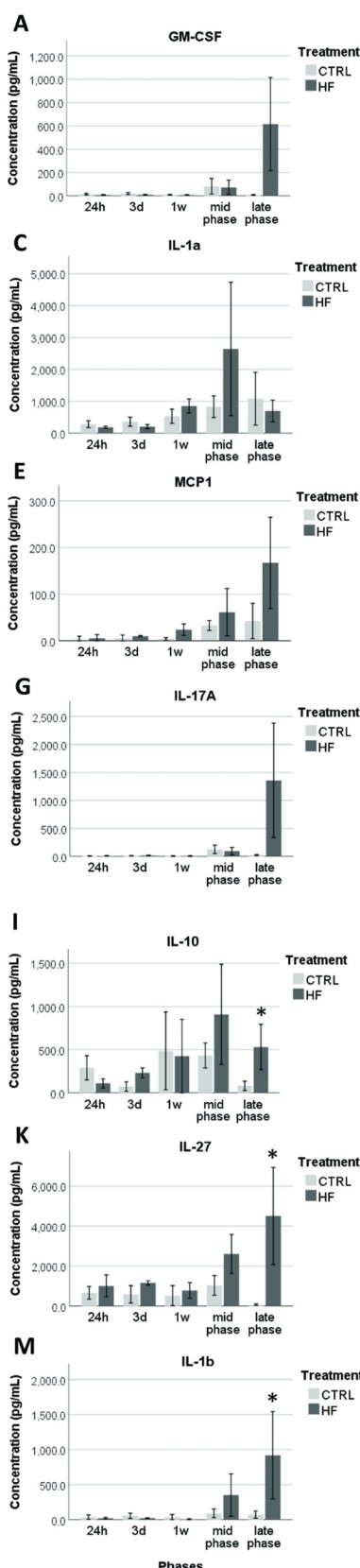

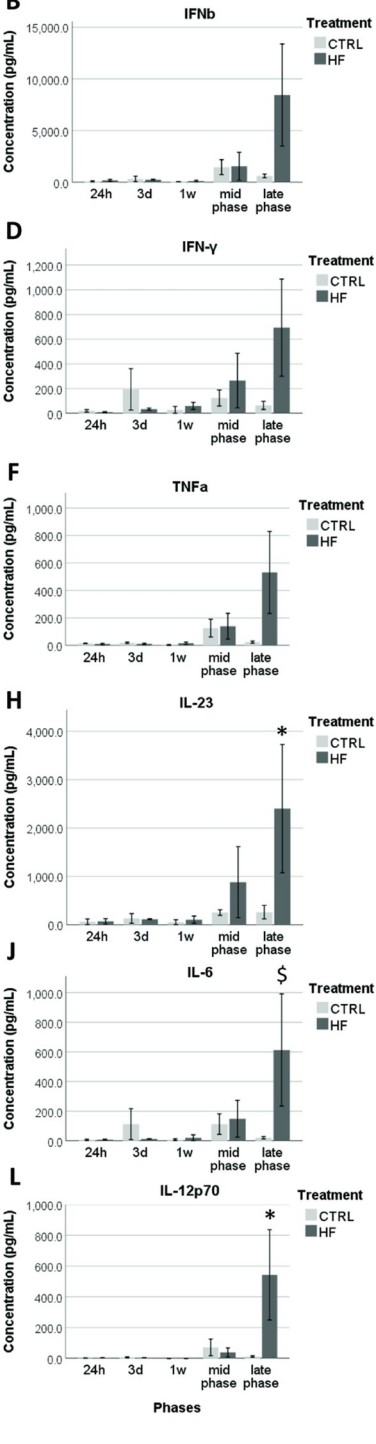

**Fig 3. HF/HS diet effects on serum concentrations (pg/mL) of different cytokines in Swiss mice.** GM-CSF (A), IFN-β (B), IL-1α (C), IFN-γ (D), MCP-1 (E), TNF-α (F), IL-17A (G), IL-23 (H), IL-10 (I), IL-6 (J), IL-27 (K), IL12p70 (L), IL-1β (M). Data are shown as mean ± SEM. Significant differences ($P \leq 0.05$) between HF/HS and control groups per time point are indicated by an asterisk (*). Tendencies ($P < 0.1$ and $> 0.05$) are indicated by a dollar sign ($).

were found for *NRF1* (Fig 5D). As was observed for *PRDX1*, *NRF2*, which is also related to OS, only showed a significantly increased expression level after 3d ($P = 0.021$) (Fig 5E). *SOD2* expression was significantly increased in the HF/HS group during the mid-phase ($P = 0.042$). No significant differences were observed at other time points (Fig 5F). *BiP* and *ATF4* were analyzed to evaluate ER-stress, and both were significantly higher in the HF/HS group 3d after the start of experimental feeding with P-values of 0.006 and 0.008 respectively (Fig 5G and 5H). *IL-1β* was significantly higher expressed in the HF/HS group during the mid-phase ($P = 0.002$) (Fig 5I). After 3d, *HSPD1*, a marker gene for mitochondrial stress, was significantly upregulated in the HF/HS group ($P = 0.008$). *HSPE1* never showed significant different expression levels (Fig 5J and 5K). Finally, *HSPA8*, which is related to protein folding, was significantly increased in the HF/HS group after 3d with a P-value of 0.002 (Fig 5L). The graphs (Fig 5) show that, all genes (except *HSPD1*, *NRF2* and *ATF4*) were downregulated in the HF/HS compared to the controls at 1w and during the late phase. However, these differences were not significant. A positive correlation was found between *BiP* and *ATF4* (r = 0.665; $P = 0.000$) and between *NRF1* and *NRF2* (r = 0.741; $P = 0.000$) (S4 Table).

## Discussion

The aim of this study was to investigate the effect of feeding a HF/HS diet for different durations (from 24h to 16 weeks) on the oviductal epithelial cell (OECs) physiology in both B6 (inbred) and Swiss (outbred) mice. We chose to focus this investigation at the transcriptomic level because changing gene expression is the first cellular response to stress. We focused the analysis on a strategically selected list of genes that are known to be involved in cellular lipotoxicity and oxidative stress. This maximises the chance of detecting specific changes compared to wide screening techniques such as RNAseq where such differences might be overlooked. Feeding a HF/HS diet resulted in a significant increase in body weight and blood cholesterol concentrations compared to the controls very early after starting the HF/HS diet feeding. These effects progressively increased over time and were later, in the Swiss mice, associated with a significant increase in circulating inflammatory cytokines after 12w compared with mice fed a control diet. Feeding a HF/HS diet altered the expression of genes related to oxidative stress (OS) in the OECs as early as 3d, which was subsequently followed by a cascade of transcriptomic changes related to mitochondrial reactive oxygen species (ROS) production and ER-stress, and ultimately transcriptomic changes showing a local inflammatory response during the late phase (12-16w). The magnitude and timing of these transcriptomic changes were strain dependent.

### Temporal systemic metabolic changes after the introduction of a HF/HS diet

The difference in weight gain between HF/HS and controls mice was significant starting from 1w of feeding in the Swiss mice and from 2w in the B6 mice, which is in line with our previous report [17]. The response of Swiss mice in weight gain was more variable compared to B6 mice due to their outbred nature. Swiss mice reached a maximum weight (plateau) at 12w whereas the B6 continued to increase in weight until week 16. This might be due to a metabolic adaptation to the increased adiposity in older Swiss mice [47] while the B6 mice appear to maintain

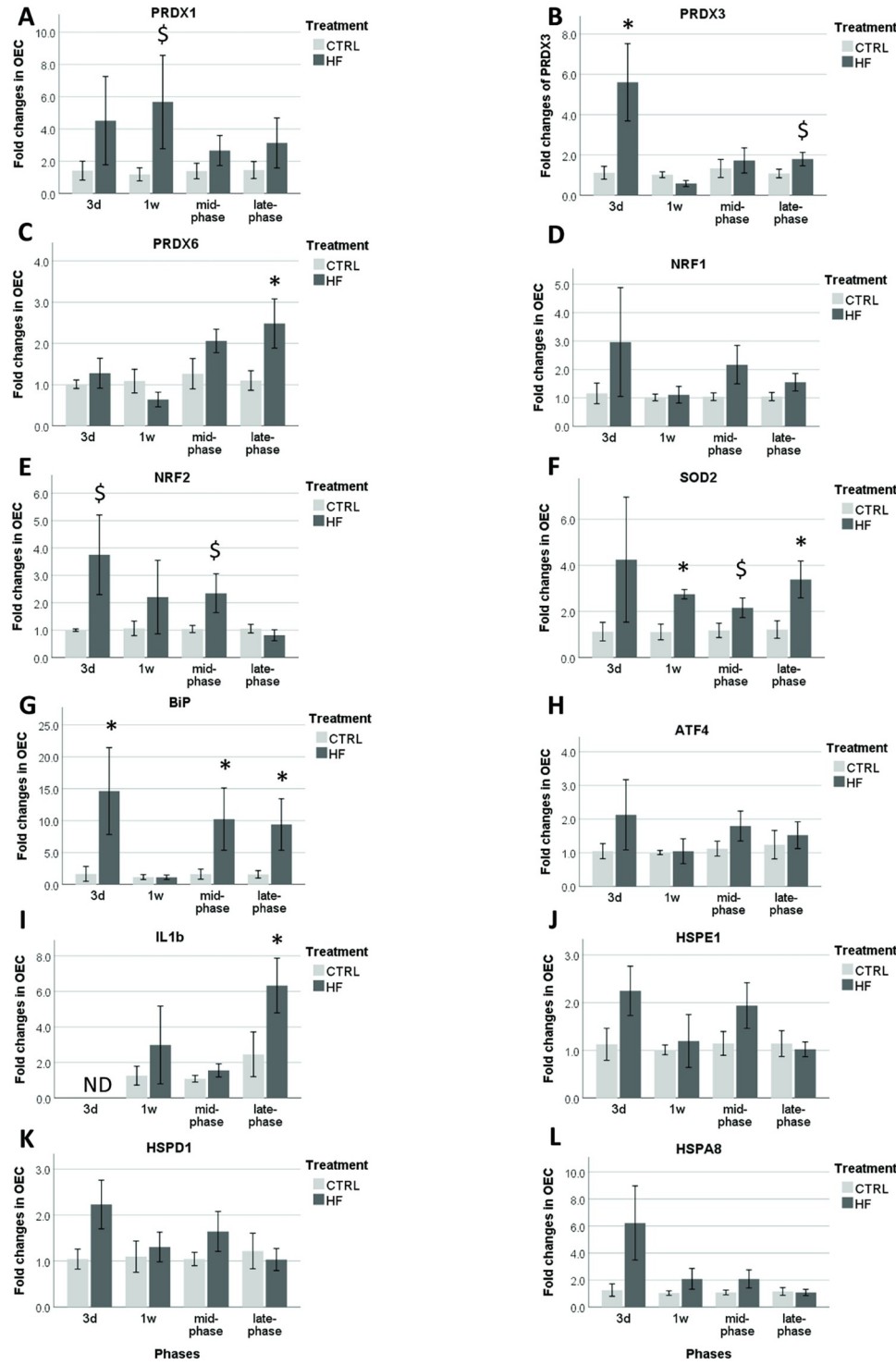

**Fig 4.** HF/HS diet effects on transcription markers of oxidative stress (A-F), ER-stress (G, H), inflammation (I), mitochondrial stress (J, K) and protein folding (L) in Swiss OECs. Columns display mean ± SEM of fold changes relative to housekeeping genes. Significant changes ($P \leq 0.05$) between HF/HS and control groups per time point are indicated with an asterisk (*). Tendencies ($P < 0.1$ and $> 0.05$) are indicated by a dollar sign ($). ND stands for 'not detected'.

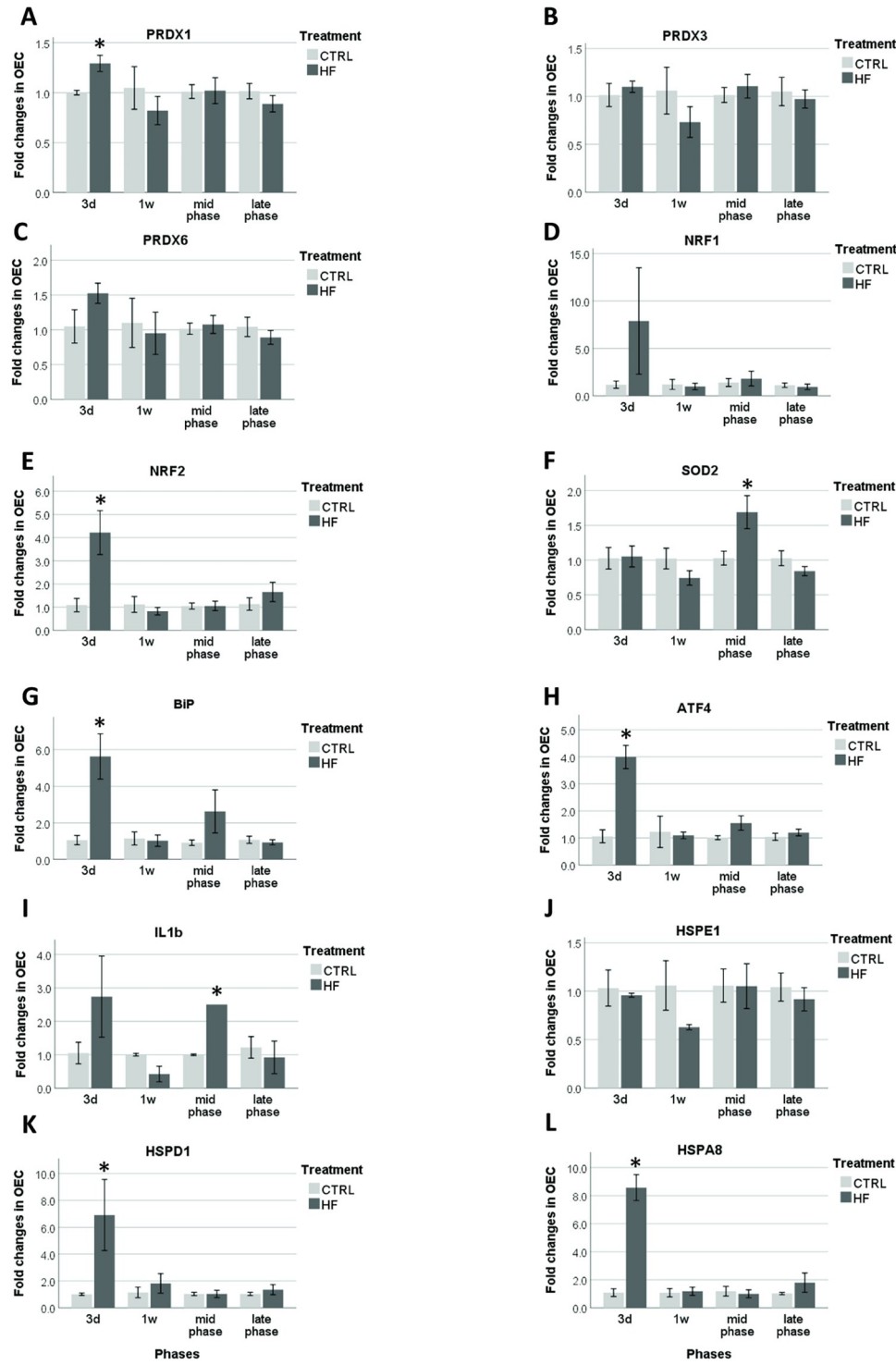

**Fig 5.** HF/HS diet effects on transcription markers of oxidative stress (A-F), ER-stress (G, H), inflammation (I), mitochondrial stress (J, K) and protein folding (L) in B6 OECs. Columns display means ± SEM of fold changes relative to housekeeping genes. Significant changes ($P \leq 0.05$) between HF/HS and control groups per time point are indicated with an asterisk (*).

their ability to store fat in their adipocytes [48]. Feeding a HF/HS diet also led to a significant increase in serum total cholesterol concentration both in Swiss (from 1w onwards) and B6 (from 3d onwards) mice. The significant increase in body weight and serum total cholesterol concentrations confirm that not only the B6 [49], but also the Swiss strain is metabolically sensitive to the high dietary saturated fat and fructose content [17]. Such acute increase in serum cholesterol in B6, already from 3d after feeding, was also reported by Williams, Campbell [38]. Cholesterol concentrations continued to increase over time in both HF/HS groups and were also more variable in the Swiss mice. Total cholesterol concentrations were relatively lower in the B6 compared to the Swiss mice regardless of the diet, which is obviously due to the genetic background [50]. Different factors might influence the difference in total cholesterol concentrations, such as increased hepatic synthesis rates of cholesterol due to obesity [51] or intestinal cholesterol absorption which showed to have a direct correlation to serum cholesterol concentrations in mice [52]. These factors may suggest a lower capacity of B6 mice to synthesize cholesterol in the liver or absorb cholesterol. Our data also show a positive correlation between body weight and total serum cholesterol concentrations, which was expected since increased weight gain has been shown to be associated with a higher rate of cholesterol synthesis [53].

To investigate if a HF/HS diet can also induce a systemic inflammatory response, and to study the time by which such response is detected after feeding, we analyzed the 13 most commonly investigated inflammatory cytokines in serum using a multiplex immunoassay. We found that six of the tested cytokines, namely IL-6, IL-10, IL12p70, IL-23, IL-27 and IL-1β, were significantly higher during the late phase (12-16w) in the HF/HS mice serum compared to the controls, whereas no signs of inflammation could be detected earlier. Williams, Campbell [38] reported an increased systemic inflammation in B6 mice (increased serum IL-6) only after 12w of HFD feeding, but an acute inflammatory response was also observed at 3d. IL-6 is produced by adipocytes and therefore its increased concentration might be due to obesity [54, 55]. On the other hand, our data show that the other measured cytokines including TNF-α and MCP-1 were not changed at any time point, which is also in line with previous studies [3, 38, 56]. Surprisingly, we found that IL-10, a cytokine with an anti-inflammatory capacity that limits the production of IL-1α, IL-1β, IL-6, IL-12, TNF-α and MCP-1 [57] was increased during the late phase in the HF/HS group. This increase was not detected in the study of Li, Wu [56] in Kunming, C57BL/6, BALB/c and ICR mouse strains and may be compensatory. IL-6, a pro-inflammatory cytokine which is stimulated by IL-17A [58] and IL12p70, a pro-inflammatory cytokine related to IL-27 and IL-23 [59] were still significantly higher or tended to be higher in the HF/HS group during late phase. It is probably not a coincidence that also IL-27, a cytokine with both pro-inflammatory and anti-inflammatory capacities and IL-23, which is considered a pro-inflammatory cytokine, were significantly increased during this phase [60, 61]. Besides IL-6, IL-17A is also known to stimulate the production of IL-1β, which was also increased during late phase in the serum of HF/HS mice [58].

## Acute and long-term effects of dietary induced metabolic stress on the oviduct

As mentioned in the introduction, the impact of HF/HS diet-induced metabolic alterations on the oviductal cell functions are not clearly defined. As described in other cell types, the mechanisms by which hyperlipidaemia induces lipotoxicity are known to be mainly mediated through OS, mitochondrial dysfunction, ER-stress and local cellular inflammatory responses [11, 13, 42]. Therefore we focused on these pathways in our OEC gene expression analysis. Studies for example showed an increase in proinflammatory genes and genes related to OS in ovaries of 4w HF-fed Sprague Dawley (SD) rats [62].

We could detect changes in the expression of some of the target genes, an effect which was dependent both on the duration of feeding and on the mouse strain. **In Swiss mice**, we observed that **3d** of feeding a HF/HS diet already resulted in increased expression of genes related to ER-stress and OS in the OECs; since both *BiP* and *PRDX3* were significantly upregulated. BiP is an ER chaperon protein that binds to misfolded proteins in the ER lumen and helps stabilizing and restoring the efficiency of protein folding by initiating UPRs [63]. Nevertheless, *ATF4*, the downstream transcription factor of *BiP*, was not upregulated at any of the time points, meaning that the ATF4-mediated UPR in ER was not initiated. PRDX3 is a mitochondrial AO [64] and hence its upregulation indicates an increased cellular OS level particularly within the mitochondria [65]. These changes were accompanied by a tendency for a higher *NRF2* expression also at 3d, a protein that reacts to OS and ER-stress and act as a transcription factor that upregulates the expression of genes harbouring the antioxidant response element (ARE) sequence in their promotor, like the mitochondrial *SOD2* [66–68]. Altogether, this suggests an acute increase in mitochondrial OS and (mild) ER-stress in response to introducing a HF/HS diet, even before a significant change in body weight or serum cholesterol could be detected. On the other hand, the expression of NRF1, which is a transcription factor with a similar binding specificity and expression profile as NRF2 [67] was not significantly affected by diet. Nevertheless, the level of expression of NRF1 and NRF2 were significantly positively correlated in the Swiss OEC data. A few days later, the increase in *NRF2* expression observed at 3d was indeed followed by an upregulation of *SOD2* at 1w (early phase), showing that the downstream effect of *NRF2* on *SOD2* was provoked [69]. An increase in *SOD2* expression was also previously recorded in the white adipose tissue of obese mice and in the uterus, but not in the ovaries, of Wistar rats that were fed a HFD for 8w [65, 68]. In addition, Jordaens et al. (2017) reported a significant increase in *SOD1* expression in bovine OECs that were exposed to lipotoxic concentrations of free fatty acids *in vitro* [70]. Nevertheless, according to our knowledge, our data here is the first to report such early NRF2-SOD2 OS response in the oviduct. The increase in *SOD2* reported here was also accompanied by a tendency for a higher *PRDX1* (AO) expression, which is also considered an NRF2-regulated gene [71]. At the ER level, and unlike the response at 3d, *BiP* expression in the HF/HS group at 1w was similar to the controls indicating that protein folding in the ER was temporarily stabilized, and confirming that the ER-stress exhibited at 3d was mild. During the mid-phase (4-8w) a second wave of *NRF2/SOD2* AO response and ER-stress (*BiP*) was apparent. But again, no significant change in the *ATF4* expression was detected. Finally, in the late phase (12-16w), in addition to the *SOD2* and *BiP* which remained upregulated in the HF/HS OECs, a significantly higher expression of both *PRDX6* and *IL-1β* was detected. This timing of PRDX6 upregulation is similar to that reported in Swiss oocytes after 13w of feeding a high fat diet [17]. PRDX6 is an AO enzyme modulated by the inflammatory cytokines IFN-γ and TNF-α. It regulates TNF or IFN-induced apoptosis through IL-1β production, by its phospholipase A2 (PLA2) activity [72–75]. Hence, the simultaneous upregulation of *PRDX6* and *IL-1β* strongly suggests a chronic diet-induced inflammatory response in the oviduct during the late phase. PRDX6 does not only play a protective AO role, but is also considered as an activator of inflammatory pathways [75].

It is important to note the dynamics of the different PRDX antioxidants (PRDX1, 3 and 6) in the OECs in this study in response to the HF/HS diet. Unlike PRDX6, both PRDX1 and PRDX3 are not modulated by cytokines. And while PRDX1 and PRDX6 are localized in the cytosol, PRDX3 is expressed in the mitochondria [72, 73]. Therefore, the early upregulation of *PRDX3* at 3d may indicate that mitochondria are more sensitive to lipotoxicity compared to the ER in the OECs. Nevertheless, markers of mitochondrial stress (*HSPD1*, *HSPE1*) were not affected by the HF/HS diet. However, it has been previously shown that feeding a high-fat diet for 16 weeks induces mtDNA damage which correlates with mitochondrial dysfunction and

increased OS in skeletal muscle and liver of mice [76]. Again, in our study this indicates that the level of mitochondrial OS was not strong enough to induce mitochondrial UPRs in the OECs.

On the other hand, **in B6 mice**, the effects of the HF/HS diet on OECs gene expression were similar but not identical to the responses seen in the Swiss mice. Likewise, the changes in gene expression suggested an increase in OS and ER-stress already at 3d after the start of the HF/HS diet feeding. *NRF2* (AO), *PRDX1* (AO) and *BiP* (ER-stress) were significantly upregulated at 3d in the HF/HS B6 OECs. The earlier upregulation of NRF2 downstream gene *PRDX1* compared to the Swiss OECs (where it was upregulated at 1w) may be explained by the fact that NRF2 is not only activated by increasing levels of ROS, but also by increased circulating cholesterol as observed by Ma [71], since serum cholesterol concentrations were increased earlier in B6 mice compared to Swiss mice. In contrast to the Swiss OECs, *ATF4* (the downstream transcription factor of BiP) was also significantly increased after 3d. This indicates an early activation of the ATF4-dependent UPRs in the ER and a higher level of cellular stress compared to that experienced in the Swiss OECs [77]. *BiP* and *ATF4* expression levels were positively correlated in the B6 OEC samples (r = 0.665; $P < 0.001$) but not in Swiss. *HSPA8*, which is important for the correct folding of proteins, was also significantly increased during the acute phase (3d). This is different from the Swiss mice where no changes in *HSPA8* expression were recorded, again suggesting a higher degree of stress in the B6 mice. As observed in the Swiss strain, *NRF1* expression was positively correlated with *NRF2* but was not significantly upregulated. It is remarkable that *HSPD1*, which is a marker gene for mitochondrial stress and responsible for refolding of misfolded proteins in the mitochondria [78] was also significantly upregulated at 3d. This indicates that, unlike in Swiss mice, the HF/HS diet induced a very acute mitochondrial stress in B6 OECs. These cells failed to upregulate PRDX3 expression which may have contributed to increased mitochondrial stress levels. Marei et al. (2020) also reported an increase in mitochondrial stress in cumulus cells of HFD fed B6 mice that was not found in oocytes or cumulus cells of HFD fed Swiss mice. However, in their study it was *HSPE1* that was significantly differentially expressed [17]. Such higher mitochondrial sensitivity in B6 has been previously described to be genetic in origin due to inbreeding. Cross insertion of mtDNA from B6 mice to the outbred C3H/HeN mice *in vivo* resulted in higher mitochondrial inner membrane potential and the generation of more ROS in cardiomyocytes [79]. A higher proportion of abnormal mitochondria was also detected in the oocytes of the B6 compared to the Swiss oocytes even in mice fed a normal diet [17]. Subsequently, and in contrast to the changes observed at 3d, none of the tested genes was significantly affected by the diet during the early phase (at 1w), suggesting that the stress levels were temporarily normalized or stabilized by the earlier adaptations in gene transcription. After longer exposure to the diet, during the mid-phase, a significant increase in mitochondrial *SOD2* was detected, showing a second wave of mitochondrial OS and the presence of a *NRF2/SOD2* AO response. In addition, *IL-1β* was also significantly increased during this phase, indicating a relatively earlier inflammatory response compared to that seen in the Swiss OECs at the late phase. However, this change was not accompanied with changes in *PRDX6* expression in B6. Upregulation of *IL-1β* was also reported in the muscles of B6 mice after 12w of HFD feeding [38]. This relatively earlier sign of cellular inflammation in B6 compared to Swiss might also be due to the relatively higher cellular stress levels. Finally, during the late phase, none of the tested genes was significantly affected by diet despite the continued increase in weight gain and hypercholesterolemia. This could be due to the advanced age of the mice at this time point and the relatively shorter reproductive lifespan of the B6 compared to Swiss mice [80].

Our gene expression data shows that the sequence of pathophysiological changes that are known to occur at the cellular level in response to diet-induced lipotoxicity are evident in the

oviduct of both Swiss and B6 mice. In other words, the OECs appear to be sensitive to the maternal metabolic stress induced by the HF/HS diet. These cells are in direct contact with the embryo and create its micro-environment [31]. Based on our in-depth analysis of the strategically-selected target genes, we can state that the sequence of changes in Swiss mice start with an acute increase in OS and protein misfolding in the ER followed by the initiation of a cascade of transcriptomic changes to control mitochondrial ROS production and ER-stress. The specific metabolic changes in B6 mice are mainly observed during the acute phase (3d) and involve higher levels of mitochondrial OS. The level of cellular stress appears to be higher in B6 to the level that stimulates the expression of UPR genes in the ER and mitochondria. Despite the activation of some endogenous AO mechanisms, chronic exposure to such stress results in local inflammatory responses that were initiated earlier in B6 mice compared to the Swiss mice.

It is important to notice that these very interesting and biologically relevant differences that we could detect between the dietary groups at different timepoints validated one another as they formed a very logical network of molecular interaction and cellular events. The fact that similar pathways are affected in a similar sequence (with a few differences in severity) in both mouse strains can also be seen as a strong validation of the described responses.

We are the first to show that the impact of the HF/HS diet on OECs can lead to changes in the oviductal microenvironment that may occur as a very acute oxidative stress response to the diet even before the development of an obese phenotype. These changes can put the developing embryo at risk and may lead to reduced fertility. This illustrates that the mechanism by which the obesogenic diet impacts fertility is not only mediated through reduced oocyte quality, but might also directly impact early embryo development in the oviduct, leading to long-term effects on fetal development, pregnancy success, and postnatal health through epigenetic alterations. Therefore, our findings might explain why subtle nutritional challenges exerted exclusively during the preimplantation period, resulted in offspring with a higher risk of developing deleterious phenotypes in adulthood [34]. However, such impact on the developing zygote within the oviduct of HF/HS diet-fed mothers is practically and technically very difficult to investigate.

In **conclusion**, administration of a HF/HS diet for a short period (as shown here at 3d) results in acute systemic changes and acute local OS or mitochondrial stress effects on OECs, evident already before the development of an obese phenotype. The acute effects in the OECs initiate a cascade of transcriptomic changes to control mitochondrial ROS production and ER-stress. However, in the mid and late phase a persistent upregulation of (mitochondrial) OS and ER-stress is observed, with ultimate signs of local and systemic inflammation in the late phase.

To the best of our knowledge, this is the first study that describes the effect of metabolic stress on the oviductal microenvironment *in vivo* and illustrates that oviductal cells can sense (or can react on) systemic diet-induced hyperlipidaemia. Further studies in our laboratory currently focus on the impact of such increased OEC stress levels on early embryo development. This study also shows different responses to a HF/HS diet between Swiss and B6 mice. The acute responses showed a higher mitochondrial sensitivity in B6 mice compared to the Swiss mice, which were showing acute OS responses. B6 mice showed earlier local inflammatory responses compared to the Swiss mice. These differences indicate that the effects of a HF/HS diet are dependent on the genetic background which should aid in designing further research.

## Supporting information

**S1 Table. Functions and full names of cytokines measured in the serum.**
(PDF)

**S2 Table. Functions and full names of genes of interest used for qPCR.**
(PDF)

**S3 Table. Pearson correlation Swiss mice.**
(PDF)

**S4 Table. Pearson correlation B6 mice.**
(PDF)

## Acknowledgments

The authors acknowledge Silke Andries, Els Merckx and Sara Verheyen for their outstanding technical assistance with animal handling, sample collection and analysis.

## Author Contributions

**Conceptualization:** Kerlijne Moorkens, Jo L. M. R. Leroy, Waleed F. A. Marei.

**Data curation:** Kerlijne Moorkens, Waleed F. A. Marei.

**Formal analysis:** Kerlijne Moorkens, Sara Verheyen, Waleed F. A. Marei.

**Funding acquisition:** Jo L. M. R. Leroy.

**Investigation:** Kerlijne Moorkens, Sara Verheyen, Waleed F. A. Marei.

**Methodology:** Kerlijne Moorkens, Waleed F. A. Marei.

**Project administration:** Jo L. M. R. Leroy.

**Resources:** Kerlijne Moorkens.

**Software:** Kerlijne Moorkens, Waleed F. A. Marei.

**Supervision:** Jo L. M. R. Leroy, Waleed F. A. Marei.

**Validation:** Kerlijne Moorkens, Waleed F. A. Marei.

**Visualization:** Kerlijne Moorkens, Sara Verheyen, Waleed F. A. Marei.

**Writing – original draft:** Kerlijne Moorkens.

**Writing – review & editing:** Jo L. M. R. Leroy, Sara Verheyen, Waleed F. A. Marei.

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
