## [Decision Letter · Decision Letter 0]

3 Aug 2022

PONE-D-22-12367Effects of an obesogenic diet on the oviduct depend on the duration of feeding.PLOS ONE

Dear Dr. Moorkens,

Thank you for submitting your manuscript to PLOS ONE. After careful consideration, we feel that it has merit but does not fully meet PLOS ONE’s publication criteria as it currently stands. Therefore, we invite you to submit a revised version of the manuscript that addresses the points raised during the review process.

ACADEMIC EDITOR: Sorry for a delayed response. I tried to contact potential reviewers, but many of them declined to review. Please consider the comments from the reviewer. 

We look forward to receiving your revised manuscript.

Kind regards,

Yi Cao

Academic Editor

PLOS ONE

Journal Requirements:

Reviewers' comments:

Reviewer's Responses to Questions

**Comments to the Author**

1. Is the manuscript technically sound, and do the data support the conclusions?

Reviewer #1: Yes

2. Has the statistical analysis been performed appropriately and rigorously? 

Reviewer #1: Yes

3. Have the authors made all data underlying the findings in their manuscript fully available?

Reviewer #1: Yes

4. Is the manuscript presented in an intelligible fashion and written in standard English?

Reviewer #1: Yes

5. Review Comments to the Author

Reviewer #1: In this paper, Kerlijne and colleagues determined the effect of obesogenic diet on the oviduct of mice during different feed duration. They founded that HF/HS elevated ER-stress and oxidative stress genes in OECs after 3days and mitochondrial antioxidants were upregulated after one week. Serum inflammatory cytokines were increased during the late-phase. In general, the present manuscript is a superficial study with very limited data present. I have the following comments for the authors to address before further consideration.

1. All 6 figures across the manuscript can be combined to 2 or 3 figures. The schematic diagram is far away from representing as a dependent figure.

2. Was fasting preformed before sacrifice the mice?

3. All figures were present in low resolution.

4. Most of the data present in the study was from RT-PCR which will compromised the convincing of the results.

5. The inflammatory cytokines were detected in the serum. Does these cytokines were detected in the OECs?

6. PLOS authors have the option to publish the peer review history of their article (what does this mean?). If published, this will include your full peer review and any attached files.

Reviewer #1: No

---

## [Author Response · Author response to Decision Letter 0]

9 Sep 2022

We thank the reviewer for taking the time to review this manuscript. We would like to highlight that the present study is the first to investigate the impact of an obesogenic diet on the oviductal microenvironment in a time dependent manner. This was performed using strategically selected inbred and outbred mouse models which are routinely used in obesity related fertility research. This provides a unique direct comparison in cellular metabolic responses that can be influenced by strain differences in mitochondrial sensitivity. This is crucial for the extrapolation of the data to understand similar potential mechanisms in the humans. In addition, the different durations of feeding were also carefully selected to coincide with specific metabolic windows of acute and chronic exposure based on previous literature. This allowed a clear distinction of immediate responses due to direct dietary effects, and later changes observed after the development of the obese phenotype and the associated unhealthy metabolic phenotype. Last but not least, the comprehensive list of genes of interest were carefully selected to cover pathways related to the most common pathways that are known to be affected by an obesogenic diet in other cell types. Because of that, our results could provide a deep insight into a network of sequential events that could demonstrate how the oviductal epithelial cells can quicky sense metabolic stress in a matter of a few days, and how such local metabolic stress can evolve over time. We could also demonstrate important differences in cellular responses between the inbred and outbred mouse model which further highlights the importance of choosing physiologically relevant mouse models in other studies and which should aid in correct interpretation and comparisons of results of similar studies performed in other laboratories. We believe that such deep analysis and novel insights can be useful for many readers and form an important fundament for further studies that focus on protecting and enhancing embryo development during the very early stages under diet induced metabolic stressed conditions. Based on the reviewer’s comment, we have added a few sentences to the discussion to clarify these points and choices; see lines 330-334, 390-393, 418-419, 498-502, 503-505, 509-511, 520-522, 527-528 in the highlighted Revised Manuscript with track changes. 

Answer to Reviewer comment 1: We agree that the schematic diagram originally shown in figure 1 is not necessary and that it is sufficient to describe the experimental design in text. So following this remark we decided to delete figure 1. Most of the other figures are already formed of a compilation of several graphs of different strains (B6 and Swiss) or different measures (cytokines or genes). Therefore we do not think it is possible to further combine these figures together. 

Answer to Reviewer comment 2: We are sorry we did not make this clear in the text. It was not possible to fast the mice before sacrification because the experimental design include timepoints after very short periods of dietary exposure. Especially for the 24h and 3days feeding, an overnight fasting might have alleviated the acute effects that we were aiming to detect. For consistency, we decided not to fast the mice used at later timepoints otherwise we cannot make a valid comparison with the acute effects. Our focus was mainly to detect potential cellular stress in the oviductal microenvironment in response to the obesogenic diet or obesity, and therefore fasting can be seen as contradictory. This has been now explained in the main text on line 135-136.

Answer to Reviewer comment 3: Thank you for this remark. We have now uploaded high resolution images that comply with the journal standards. 

Answer to Reviewer comment 4: We agree that we mainly relied on qPCR in this study. This choice was made after careful consideration for several biological, technical and practical reasons. First, the amount of epithelial cells that can be collected from a murine oviduct is a limiting factor, which restricts the number of measurements that can be done per sample. In addition, since this is the first study to examine the impact of an obesogenic diet on the oviduct in such timely manner, it is logical to focus the analysis on the transcriptomic level because biologically, changing gene expression is one of the earliest responses to cellular stress. As mentioned above, it is also important to notice that the selected genes form a functional and sequential network of events. For example, when we detect an acute increase in NRF2 (and not NRF1) followed by an upregulation of SOD2 a few days after, this can be seen as a validation and confirmation of the results, one another. The similarities and specific differences between the two mouse models can also been seen as an important validation of the results. On the other hand, technically, qPCR is the only technique that allows the inclusion of such high number of samples on the same run, including all timepoints and replicates, to avoid bias from comparing different batches or handling. Opting for a more general analysis such as RNAseq was also considered. But we preferred to target specific pathways that are highly expected to be affected by lipotoxicity and inflammation. Based on the reviewer’s comment, we have added some emphasis on these aspects in the main text on lines 330-334 and 498-502. 

Answer to Reviewer comment 5: No, technically it was not possible to do this analysis on the oviductal cells because of the low amount of the biological material.

---

## [Editor Report · Decision Letter 1]

15 Sep 2022

Effects of an obesogenic diet on the oviduct depend on the duration of feeding.

PONE-D-22-12367R1

Dear Dr. Moorkens,

We’re pleased to inform you that your manuscript has been judged scientifically suitable for publication and will be formally accepted for publication once it meets all outstanding technical requirements.

Kind regards,

Yi Cao

Academic Editor

PLOS ONE
---

## [Editor Report · Acceptance letter]

21 Sep 2022

PONE-D-22-12367R1 

Effects of an obesogenic diet on the oviduct depend on the duration of feeding. 

Dear Dr. Moorkens:

I'm pleased to inform you that your manuscript has been deemed suitable for publication in PLOS ONE. Congratulations! Your manuscript is now with our production department. 

Kind regards, 

on behalf of

Dr. Yi Cao 

Academic Editor

PLOS ONE